# INSTRUCTION AGENT: ENHANCING AGENT WITH EXPERT DEMONSTRATION

## ABSTRACT

Graphical user interface (GUI) agents have advanced rapidly but still struggle with complex tasks involving novel UI elements, long-horizon actions, and personalized trajectories. In this work, we introduce *Instruction Agent*, a GUI agent that leverages expert demonstrations to solve such tasks, enabling completion of otherwise difficult workflows. Given a single demonstration, the agent extracts step-by-step instructions and executes them by strictly following the trajectory intended by the user, which avoids making mistakes during execution. The agent leverages the verifier and backtracker modules further to improve robustness. Both modules are critical to understand the current outcome from each action and handle unexpected interruptions(such as pop-up windows) during execution. Our experiments show that Instruction Agent achieves a 60% success rate on a set of tasks in OSWorld that all top-ranked agents failed to complete. The Instruction Agent offers a practical and extensible framework, bridging the gap between current GUI agents and reliable real-world GUI task automation.

## 1 INTRODUCTION

Graphical User Interface (GUI) agents leverage UI elements and input methods such as keyboard and mouse to interact with digital devices similarly to humans. GUI agents are predominantly powered by Multimodal Large Language Models (MLLMs). Recent research on GUI agents Koh et al. (2024); Cheng et al. (2024); Zhang et al. (2025); OpenAI (2025); Anthropic PBC (2024); Qin et al. (2025) has demonstrated significant potential in automating user interactions and facilitating diverse tasks across digital platforms, including desktops and mobile devices. Since the introduction of evaluation benchmarks such as OSWorld Xie et al. (2024) and Windows Agent Arena Bonatti et al. (2024), rapid progress has been made in the performance capabilities of GUI agents. For instance, in the OSWorld benchmark Xie et al. (2024), the task success rate has increased from 5.8% in April 2024 (achieved by Gemini Vision Pro) to 42.9% by OpenAI's CUA O3 model. Nevertheless, a substantial performance gap substantial gap remains compared to human-level performance (72.36%).

Despite the rapid development of GUI-based agents, some tasks remain difficult or infeasible for current agents due to complex UI elements, highly idiosyncratic procedures, or long-horizon workflows spanning many steps. However, automating these tasks is still highly desirable.

In this paper, we present a method that automates many tasks previously unsolved by the SOTA agents using a single human demonstration. Our agent is a test-time–only, training-free system built on existing LLM APIs and open-source models. From the demonstration, it derives a high-quality task plan as well as UI grounding hints for the grounding model. To ensure the correctness and reliability of each action execution, we added verification and backtracking modules. Specifically, our framework comprises the following components (Figure 1):

1. An *Instructor Model* that extracts precise action plans from user demonstrations. 2. An *Actor Model* that follows these demonstration plans step-by-step, with built-in error tolerance and uncertainty handling modules.

Although our framework relies on expert demonstrations, these demonstrations can be easily captured through quick, ad-hoc user recordings. Our agent framework is particularly beneficial in the following scenarios:

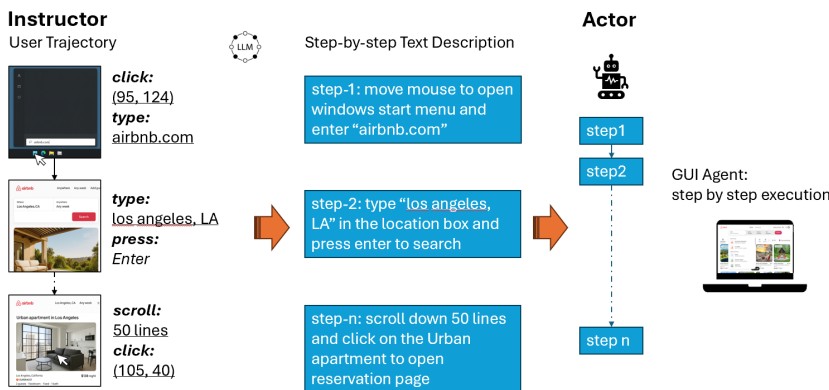

Figure 1: Instructor–Actor Agent

1. *Novel or unintuitive UI elements*: Applications frequently introduce novel UI elements that even humans find challenging to understand. Figure 2 provides an example such of UI elements. Humans typically learn from interacting with these elements, and this knowledge can be effectively transferred to agents via expert demonstrations at a minimal cost.

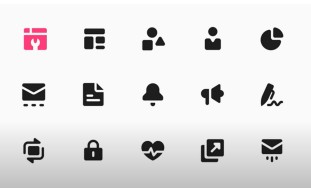

Figure 2: An example of a non-intuitive UI AxureBoutique (2023).

2. *Long-horizon tasks requiring high accuracy at each step*: For tasks involving many sequential steps, assuming the success of each step is independent: The probability of overall success ($P_{\text{success}}$) decreases exponentially with each individual step's success probability ($p_i$) :

$$P_{\text{success}} = \prod_{i=1}^{n} p_i$$

Therefore, ensuring high precision in each step is critical. Our agent's strictly following the steps from the demonstration significantly enhances the likelihood of success for individual steps and for the entire long-horizon task.

3. *Tasks requiring highly specific steps*: While many computer tasks are, in principle, solvable in multiple ways, user-specific configurations can cause different approaches to yield different outcomes. A personalized solution can be sometimes optimal for agent execution. For example, a user may prefer Firefox because it stores their credentials and browsing history; in such a setup, successfully placing an Amazon order may only be possible using Firefox with the default profile. Understanding these requirements beforehand are challenging for current agents, which tend to select the most common actions by default. Our demonstration-based approach enables the agent to exactly follow the user's personalized trajectory and avoids the bias toward "common" approaches.

4. *Tasks that are hard to express concisely*. User often specify one's task with a few sentence for GUI agent to execute. However, some tasks are difficult to describe in a short passage—for example, those with personalized, step-dependent details or long-horizon procedures. Although one could enumerate all particulars in a longer description, the agent can still miss details during execution. In such cases, a short demonstration recording is often the most reliable way to convey exactly what the user needs.

Finally, the instruction agent frame work we propose can have broader use cases. Expert demonstrations can potentially be augmented and generalized to support similar tasks. Furthermore, demonstrations and actions can be encapsulated as reusable tools or APIs, thereby enabling broader GUI task automation.

## 2 RELATED WORK

**Script-Based Automation**   Before the emergence of LLM-powered GUI agents, script-based automation methods were widely used to perform automatic computer operations Tupsakhare (2019); Oksanen (2023). These scripts, however, rely heavily on static environments and tend to fail in dynamic contexts commonly encountered in modern apps and webpages. For example, script-based solutions break easily if UI elements move due to software updates, window resizing, or unexpected pop-up windows.

**GUI Agents**   Significant progress has been made on GUI agents OpenAI (2025); Agashe et al. (2025); Anthropic PBC (2024); Bonatti et al. (2024). Early agent systems typically relied on structured representations (e.g., accessibility trees or HTML), whereas more recent work operates directly on visual screenshots. Architectures range from end-to-end models to modular pipelines with separate planning, grounding, and execution components. Despite these advances, substantial gaps remain relative to human-level performance, especially on complex or novel tasks with challenging interfaces or long-horizon trajectories.

Typical GUI-agent pipelines comprise four primary modulesWang et al. (2025): (1) Planning, which produces a step-by-step solution; (2) GUI Understanding (grounding), which identifies UI elements and their interactive affordances; (3) Action Decision, which selects concrete actions based on the plan; and (4) Execution, which interacts with the operating system via APIs. Among these, planning and grounding are reported to be the predominant sources of error: according to Agent-S2Agashe et al. (2025), planning accounts for approximately 41% and grounding for about 20.5% of failures on OSWorld.

In this paper, we directly address these two bottlenecks. The human demonstration provides high-quality plan for the agent, and instruction generation provides rich and detailed hints to improve grounding accuracy.

**GUI Grounding**   GUI grounding is a critical component in GUI-based agents, as it connects the UI environment to the agent's internal knowledge. Numerous studies have advanced this field, including UI-Tars Qin et al. (2025), U-Ground Gou et al. (2025), WinClick Hui et al. (2025), and OS-Atlas Wu et al. (2024). As of the time of writing, UI-Tars represents the state-of-the-art open-source grounding model across multiple benchmarks. Given its strong empirical performance, we adopt UI-Tars as the grounding model for our agent.

**Human-in-the-Loop**   Many work has explored introducing human interaction into agentic workflows to improve robustness, performance, and reliability Mozannar et al. (2025); Huq et al. (2025). However, these designs require human input during task execution, which is less convenient. In contrast, our agent only needs a pre-recorded demonstration as input and can execute the task automatically without human intervention.

**Demo-Based Agent Learning**   Recently, there has been increased interest in leveraging human-generated computer usage trajectories to enhance agent performance. Most existing works focus on utilizing large-scale collections of high-quality trajectories for agent training, demonstrating significant improvements in effectiveness. Examples include Synatra Ou et al. (2024) and AgentTrek Xu et al. (2025), where trajectories are extracted and refined from online textual tutorials (e.g., Wiki-How[1]) and screenshot databases (e.g., ClueWeb Overwijk et al. (2022)). Alternatively, approaches such as those in Agente He et al. (2025), OS-Genesis Sun et al. (2025), and NNetNav Murty et al. (2025) utilize exploratory agents to autonomously discover trajectories, later identifying and annotating meaningful task-oriented sequences for training.

---

[1]https://www.wikihow.com/Main-Page

However, to the best of our knowledge, all prior works rely on large-scale trajectory datasets for model training. In Li et al. (2024), it has been found that simply scaling training trajectories is not sufficient for agent to generalize for out-of-domain tasks. On the other hand, there has been research that studies directly using expert demo data for agent planning, both Jang et al. (2025) and Ruoss et al. (2025) has shown that it is ineffective to plug in expert demonstration directly in test time for agents using LLM APIs. Our approach is the first to use test-time inference based solely on expert demonstrations, requiring only a single trajectory per task without the need for extensive trajectory datasets or additional training, and has shown effectiveness. This design empowers end-users to directly create demonstrations and deploy our agent without heavy computational resources or training expertise.

## 3 METHODOLOGY

### 3.1 PROBLEM FORMULATION

We model the autonomous digital agent as a solution to a partially observable Markov decision process (POMDP), following the definitions in Xie et al. (2024) and Agashe et al. (2025). A POMDP is defined as $M = (\mathcal{S}, \mathcal{O}, \mathcal{A}, T, R)$, where:

- $\mathcal{S}$ is the state space,
- $\mathcal{O}$ is the observation space,
- $\mathcal{A}$ is the action space,
- $T : \mathcal{S} \times \mathcal{A} \to \mathcal{S}$ is the transition function, and
- $R : \mathcal{S} \times \mathcal{A} \to \mathbb{R}$ is the reward function.

In our context, the state space $\mathcal{S}$ corresponds to the digital device's state, which may include the desktop environment, webpage, or application status. The observation space $\mathcal{O}$ typically consists of screenshots representing each state. In addition, we initialize the observation $O_0$ with step-by-step instructions generated from human demonstration. We name this module as the Instructor. The reward function $R$ measures task success, i.e., it indicates whether the agent has successfully completed the assigned task.

Similarly, we formalize the instructor's task. The instructor receives a sequence of $(\mathcal{A}, \mathcal{S})$ pairs and must convert this sequence into a natural language description, which serves as part of the initial observation $O_0$ for the agent.

### 3.2 AGENT ARCHITECTURE

The overall agent architecture is composed of two modules Instructor and Actor as shown in Figure: 1. The instructor uses human recorded trajectory and output a step-by-step instruction, The actor then follows the step by step instruction and carry out each step for the same task. The Actor is further composed of Verifier, Grounder, Executor and Backtracker. In Instructor, Actor, Verifier and Executor, we use GPT-4o as the backend LLM. In the grounding module, we use UI-Tars 1.5 to get the grounding coordinates. All LLMs usage happens at test time only and there are no training involved.

#### 3.2.1 INSTRUCTOR

The instructor module consists of two main components: the **Recorder** and the **Instruction Generator** as show in Figure: 4. Our setup requires high-quality trajectory recordings, especially precise state-action pairs. In our framework, the state is represented by screenshots—we do not use accessibility trees or HTML. For input actions, we capture only keyboard and mouse inputs.

Although screen recording videos and tutorials are widely available online, most cannot be directly leveraged for instruction generation, as they typically do not capture the user's input events. Even when user inputs are visible in some recordings (such as those demo videos provided by Xie et al. (2024)), these inputs are usually not precisely aligned with the corresponding states which is the screenshot at the exact time point, making it unusable for our purpose. Therefore, an ad-hoc human

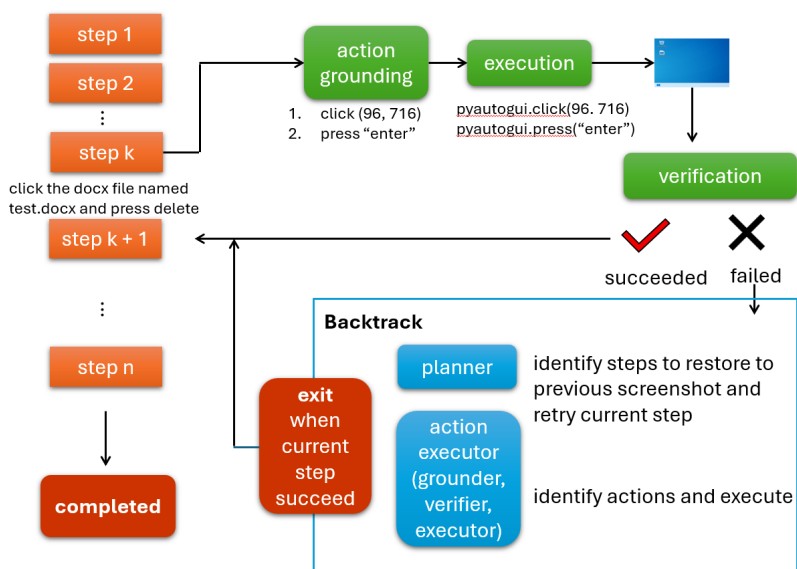

Figure 3: An overview of Instruction Agent

demonstration is still required for the agent. For each user action during the recording phase, we capture the screenshot immediately preceding that action.

During the instruction generation phase, we call a LLM, with the input consists of the user action log and its associated screenshot and prompt it to generate the a natural language description of the user action. For click actions, we annotate the input coordinates on the screenshot to enable the language model to generate more precise, location-aware descriptions, which are critical for grounding model in the actor agent. Additionally, we found it beneficial to include the screenshot after the action at the same time, as it provides useful functional feedback for describing the action's effect.

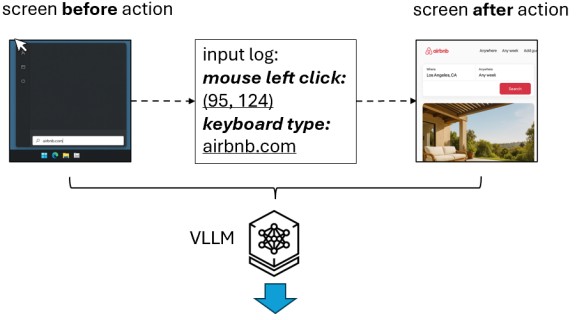

Figure 4: Instruction Generation

### 3.2.2 ACTOR

The actor module is responsible for interacting with the environment and executing the instructions provided by the instructor. It is composed of four main components: the **Grounder**, the **Verifier**, the **Backtracker**, and the **Executor**.

Given an instruction list consisting of steps $s_1, s_2, \ldots, s_n$, where each $s_i$ is an action description recorded in the demonstration trajectory, the actor iterates through the instruction list as follows:

For each step $s_i$, the agent first uses the UI Grounder to generate a specific command (e.g., click, type, scroll) based on the instruction. This command is then sent to the Executor, which converts it into executable code (i.e., using PyAutoGUI) to interact with the environment. After execution, the Verifier compares the screenshots before and after the action to determine whether the step was successfully performed. If the verifier confirms success, the agent proceeds to the next instruction; otherwise, it enters the backtracking loop to retry or recover from the failed action.

**UI Grounder:** Our agent is compatible with open-source grounding models such as U-Ground Gou et al. (2025) and UI-Tars Qin et al. (2025). In our experiment, we use UI-Tars 1.5 7B[2]. Although UI-Tars is capable of directly generating end to end executable code, we found relying on its executable code does not get satisfactory performance in our use case. Instead, we only use the UI coordinates it identifies and use a separate LLM(GPT-4o) for code generation.

**Verifier:** The verifier is a critical component of our agent. Because the agent must follow instructions step by step, it needs to know when a step has been successfully completed and it is safe to proceed, versus when a step has failed and should be retried. Without a verifier, the agent would execute steps blindly and can't complete any task when any step fails.

The verifier utilizes a LLM (GPT-4o) to determine whether an action was successful. It takes as input the screenshots before ($O_{i-1}$) and after ($O_i$) the action. By analyzing changes in the observation, the verifier decides when to proceed based on if the intended action had the desired effect.

**Backtracker:** When the verifier determines that an action has failed, the backtracker is invoked to retry the step. Given the dynamic nature of computer environments, it is often necessary to restore the previous state before retrying. The backtracker consists of a planner and an action executor (use the same LLMs as the Actor agent). It stores the observation (screenshot) prior to $s_i$ and plans a sequence of actions to return to that state (see the Backtrack module in Figure 1). In real-world environments, a reset button or snapshot may be available, but often, recovery requires additional actions such as "click back" in a browser or closing unintended pop-ups. The backtracker analyzes the current and target screenshots to make a recovery plan, then executes the actions, verifying after each step whether the agent has returned to the desired state. To avoid infinite loops, we limit the number of recovery attempts; if recovery fails after a few steps, the workflow is terminated or human intervention is requested.

While it may appear paradoxical that the backtracker itself is an autonomous agent (posing a potentially harder problem), we argue that, in practice, divergence caused by simple misclicks is usually minor. Previous research Xie et al. (2024); Koh et al. (2024) shows agents perform well in such short, localized recovery tasks. As a safeguard, the backtracker is restricted to a small number of retries before terminate the task as failure.

**Executor:** The executor module, using a large language model (GPT-4o), receives the command generated by the grounder and the current state, and outputs executable code to interact with the environment. We use pyautogui as the primary interface for execution, which is a popular and convenient choice.

## 4 EXPERIMENTS

### 4.1 BENCHMARK

We evaluate our agent using the widely adopted OSWorld benchmark Xie et al. (2024) for GUI agents. For each task, we create a human demonstration trajectory and then let the agent to act autonomously.

#### 4.1.1 TASK SELECTION

Instruction Agents have access to high-quality plans derived from human demonstrations, giving them a natural advantage in task completion. Therefore, evaluating them on tasks already solved by

---

[2]https://huggingface.co/ByteDance-Seed/UI-TARS-1.5-7B

other agents is not meaningful. Instead, we focus on tasks that failed by the top-ranked agents on the OSWorld leaderboard[3]. Specifically, we selected the top three open-source agents available at the time of writing (as of May 1, 2025) and identified tasks that all three failed to complete (Table 1). These agents were ranked 3, 4, and 6, respectively. We excluded agents ranked 1 and 2, as their trajectories are not publicly available. Nevertheless, the open-source agents we selected achieve success rates comparable to the leading closed-source agents, ensuring that our evaluation remains representative.

OSWorld contains 369 tasks in total, of which 130 were failed by all three selected agents. We randomly sampled 20 of these 130 tasks 20 of these 130 tasks to balance coverage and evaluation cost for our evaluation.

### 4.1.2 RECORDING

We hired human annotators to record expert demonstrations within Docker-hosted virtual machines. The recording script captured mouse clicks, keyboard inputs, and screenshots before and after each action. An instruction generator then converted these events into step-by-step textual descriptions, which were provided to the agent.

### 4.1.3 AGENT EVALUATION

We leveraged OSWorld's Docker environment to run the tasks. Each experiment was evaluated and manually inspected to ensure accuracy.

### 4.1.4 RESULTS

Table 1 presents the performance comparison. On the 20 sampled tasks—each of which all top agents failed—our agent achieved a 60% success rate. Although not perfect, this result is worth noting given the difficulty of OSWorld tasks: humans achieve only 72% accuracy on overall OSWorld tasks. Thus, our agent demonstrates competitive performance.

| Agent | Success Rate |
|---|---|
| **Instruction Agent (ours)** | **60%** |
| Human | 72.36% |
| UI-TARS-1.5 (100 steps) - rank 3 | 0% |
| Agent S2 w/ Gemini 2.5 (50 steps) - rank 4 | 0% |
| InfantAgent (50 steps) - rank 6 | 0% |

Table 1: Success rates of different agents on the 20 unsolved OSWorld tasks.

### 4.2 ABLATION STUDIES

We further conducted ablation studies to assess the contributions of the *verifier* and *backtracker* modules (Table 2). The verifier evaluates the correctness of each executed step, while the backtracker allows the agent to recover from errors by restoring the environment to a previous state and retrying.

In our experiments across 20 tasks, 5 required at least one retry, and in 2 cases, an incorrect action pushed the environment into a different state which which required the backtracker to restore a valid state before retrying.

| Agent Variant | Success Rate |
|---|---|
| **Instruction Agent (full)** | 60% |
| - without backtracker | 45% |
| - without verifier and backtracker | 40% |

Table 2: Ablation study: contributions of the verifier and backtracker modules.

---

[3]https://os-world.github.io/

## 4.3 FAILURE ANALYSIS

In this section, we discuss the common failure modes observed during our experiments. The failures can be broadly categorized into four types: (1) grounding errors, (2) execution errors, (3) verification errors, and (4) backtracking errors.

**Grounding Errors.** In many failed tasks, the grounding model did not accurately produce the correct coordinates, even when provided with detailed instructions. Grounding models are evolving rapidly, and we expect these errors to diminish as models improve.

**Execution Errors.** A small portion of failures stemmed from incorrect Python code (via `pyautogui`) generated by the Executor, despite having correct instructions and accurate grounding coordinates. This is expected, as we relied on a general-purpose LLM(GPT-4o) with only simple prompting for code generation. We believe such errors can be mitigated by employing models specifically fine-tuned for GUI interaction or by designing more robust prompts.

**Verification Errors.** Verification errors occur when the LLM fails to correctly determine whether a step has been successfully completed. We tackle this by adding explicit descriptions of the expected outcome of each action to our instructions. For example, instead of simply stating "click the Windows Start icon," we specify "click the Windows Start icon to open the Windows Start menu." This allows the verifier to check both the action and its effect. While this approach substantially reduced verification errors, they still occur in tasks involving novel UI interactions or subtle interface changes that are difficult for the LLM to detect.

**Backtracking Errors.** When errors from the above sources occur, the agent attempts to backtrack and recover. However, backtracking remains challenging: the agent may fail to restore the environment to a valid state or may become stuck in loops. To address this, we introduced a memory buffer that stores all previous attempts and errors encountered during backtracking, and we prompt the agent to try alternative strategies when it becomes stuck. This reduces failure rates; however, backtracking still struggles when the divergence caused by an earlier error is too severe to recover.

## 5 BROADER USE CASES

The goal of the Instruction Agent is to automate complex tasks with minimal human intervention, and our experiments demonstrate its effectiveness. Beyond hard task automation, the agent has broader applications: when tasks are common or part of a larger workflow, it can be packaged as an API or tool for integration with other agents; although task-specific, its demonstrations can often be adapted to similar tasks with only minor edits; and for tasks that remain too challenging, limited human interaction can be introduced to guide the most difficult steps. Even in such cases, prerecorded demonstrations minimize the need for continuous supervision while significantly improving task performance.

## 6 CONCLUSION

In this work, we presented *Instruction Agent*, a training-free framework that leverages expert demonstrations to automate complex GUI tasks that have be failed by current agents. Instruction Agent offers a robust and extensible framework for reliable GUI automation, enabling broader adoption in both everyday workflows and specialized enterprise applications.

Future work includes improving the backtracking mechanism and testing the framework with different LLMs, particularly smaller models that can run efficiently on edge devices. We see this as a step toward more practical, reliable, and accessible GUI automation.

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

# A APPENDIX

## A.1 INSTRUCTION FORMAT

Below is the full instruction for task `a82b78bb-7fde-4cb3-94a4-035baf10bcf0`:

```
[basicstyle=,breaklines=true] "action": "Left click on the blue,
underlined hyperlink text 'https://minedojo.org' located below the
```

author affiliations near the top center of the PDF document; this action
opens the minedojo.org website in the default web browser." "action":
"Left click on the 'Team' button, which is a white rounded rectangular
tab located at the far right of the horizontal navigation bar below
the MineDojo logo; this action scrolls the page to display the Team
section." "action":  "Left click on the blue, underlined text 'Jim
(Linxi) Fan' located below the first circular profile image in the 'Team'
section; this is a hyperlink that opens the personal website of Jim
Fan in a new browser tab." "action":  "Left click on the star-shaped
bookmark icon located in the top right corner of the Chrome address
bar, which appears as an outlined star; this action opens the 'Bookmark
added' dialog for the current page." "action":  "Left click on the
dropdown menu labeled 'Bookmarks bar' in the 'Bookmark added' popup,
located in the upper right area of the Chrome window, which has a white
background and a blue outline; this opens the folder selection options
for saving the bookmark." "action":  "Left click on the 'Choose another
folder...'  option in the dropdown menu under the 'Folder' field of
the 'Bookmark added' popup, located in the upper right section of the
popup; this opens the 'Edit bookmark' window with more folder selection
and editing options." "action":  "Left click the 'New folder' button
at the bottom left of the 'Edit bookmark' dialog, which is a rounded
rectangular button with gray text and border; this action creates a new,
editable folder named 'New folder' under 'Bookmarks bar'." "action":
"TYPE 'Liked Authors'" "action":  "Left click the blue 'Save' button
at the bottom right of the 'Edit bookmark' dialog; this button has
white text and a rounded shape.  Clicking it closes the dialog and
saves the bookmark." "action":  "Left click on the browser tab labeled
'MineDojo | Building Open...'  at the top of the Chrome window, which
is a rectangular tab element located to the left of the currently active
'Jim Fan' tab and features the MineDojo logo and page title; this action
brings the MineDojo website to the foreground, displaying the 'Team'
section." "action":  "Left click on the text link labeled 'De-An Huang'
located in the second row, third column of the 'Team' section beneath a
circular profile photo; this action opens De-An Huang's personal academic
webpage in a new browser tab." "action":  "Left click on the star-shaped
'Bookmark this tab' icon located at the right end of the Chrome address
bar, which is gray before clicking; this action opens a popup confirming
the tab is bookmarked." "action":  "Left click on the blue 'Done' button
in the 'Bookmark added' popup, located to the right side of the popup
window, to close the dialog and save the bookmark." "action":  "Left
click on the browser tab labeled 'MineDojo | Building Open-ended Embodied
Agents' at the top left of the Chrome window, which has a colorful icon
and is positioned as the first tab in the row; this action switches
the view to the MineDojo website displaying the team member profiles."
"action":  "Left click on the underlined text 'Yuke Zhu' located near
the bottom right of the team members section; it is a black link below a
circular portrait and will open Yuke Zhu's personal academic webpage in a
new browser tab." "action":  "Left click on the star-shaped bookmark icon
located at the right end of the Chrome address bar, which is outlined and
turns blue when hovered; this action opens the 'Bookmark added' popup for
the current page." "action":  "Left click the blue 'Done' button at the
bottom right of the 'Bookmark added' popup dialog in Chrome; this button
confirms adding the bookmark and closes the popup." "action":  "Left
click on the browser tab labeled 'MineDojo | Building Open...'  located
at the top left of the Chrome window, which is visually represented as
a light-colored tab with the MineDojo logo and partially visible title
text; this action brings the MineDojo team webpage to the foreground."
"action":  "Left click on the text link labeled 'Anima Anandkumar'
located at the bottom right of the 'Team' section, beneath the round

profile image with long dark hair and a light-colored top; this action
opens a new browser tab displaying the Anima AI + Science Lab webpage."
"action":  "Left click on the star icon (bookmark button) located at the
right end of the Chrome address bar, which is outlined and turns blue
when hovered; this action adds the current page to bookmarks and opens
the 'Bookmark added' dialog." "action":  "Left click on the blue 'Done'
button located at the bottom right of the 'Bookmark added' popup dialog;
this action closes the popup and returns to the main webpage."

## A.2  AGENT TRAJECTORY

Full execution trace with verifier responses for task `a82b78bb-7fde-4cb3-94a4-035baf10bcf0`:

[basicstyle=,breaklines=true] Running AgentM with run name:  1 Screenshot
received ...   ready to act

Executing step 0:  Action:  Left click on the blue, underlined hyperlink
text "https://minedojo.org" ...  Verification Response:  YES

Executing step 1:  Action:  Left click on the "Team" button ...
Verification Response:  YES

Executing step 2:  Action:  Left click on the blue, underlined text "Jim
(Linxi) Fan" ...  Verification Response:  YES

Executing step 3:  Action:  Left click on the star-shaped bookmark icon
...  Verification Response:  YES

Executing step 4:  Action:  Left click on the dropdown menu labeled
"Bookmarks bar" ...  Verification Response:  YES

Executing step 5:  Action:  Left click on the "Choose another folder..."
option ...  Verification Response:  YES

Executing step 6:  Action:  Left click the "New folder" button ...
Verification Response:  YES

Executing step 7:  Action:  TYPE "Liked Authors" (No need to verify text
input or key press)

Executing step 8:  Action:  Left click the blue "Save" button ...
Verification Response:  YES

Executing step 9:  Action:  Switch to "MineDojo | Building Open..." tab
...  Verification Response:  YES

Executing step 10:  Action:  Left click on the text link labeled "De-An
Huang" ...  Verification Response:  YES

Executing step 11:  Action:  Bookmark current tab ...  Verification
Response:  YES

Executing step 12:  Action:  Click "Done" in bookmark popup ...
Verification Response:  YES

Executing step 13:  Action:  Switch to "MineDojo | Building Open-ended
Embodied Agents" tab ...  Verification Response:  YES

Executing step 14:  Action:  Left click on the underlined text "Yuke Zhu"
...  Verification Response:  YES

Executing step 15:  Action:  Bookmark current tab ...  Verification
Response:  YES

Executing step 16:  Action:  Click "Done" in bookmark popup ...
Verification Response:  YES

Executing step 17:  Action:  Switch to "MineDojo | Building Open..." tab
...  Verification Response:  YES

```
Executing step 18:  Action:  Left click on the text link labeled "Anima
Anandkumar" ...  Verification Response:  YES

Executing step 19:  Action:  Bookmark current tab ...  Verification
Response:  YES

Executing step 20:  Action:  Click "Done" in bookmark popup ...
Verification Response:  YES

All steps completed successfully.
```

