# OpenReview forum: "Instruction Agent: Enhancing Agent with Expert Demonstration"
_ICLR.cc/2026/Conference — ICLR 2026 Conference Withdrawn Submission_

### Official Review · Reviewer_xKZj · 2025-10-26

**Soundness:** 3
**Presentation:** 2
**Contribution:** 2
**Rating:** 2
**Confidence:** 4

**Summary:**

This paper proposes Instruction Agent, a training-free GUI automation framework that uses a single expert demonstration, aiming to execute long-horizon and complex tasks. The instructor module ensures the agent follows the instruction, while the verifier and backtracker aim to ensure robustness. The approach is evaluated on OSWorld tasks that existing state-of-the-art GUI agents fail to complete.

**Strengths:**

Using expert demonstrations at test time is a practical and well-motivated direction that bridges human expertise with LLM-based agents.

**Weaknesses:**

1. Limited evaluation. The evaluation is only done on OSWorld tasks with only 1 agent type (ChatGPT-4o).
2. Limited scalability and cost: If we need a human for every new task, it would limit the scalability to hundreds of tasks or dynamic environments where agents must adapt without explicit demonstrations.
3. The verifier relies on screenshot comparison through LLM prompting, which is potentially brittle. The paper could include a quantitative metric for verifier accuracy and an ablation on prompt sensitivity for this newly introduced module.

**Questions:**

1. Generalization: Can the agent handle out-of-distribution or slightly modified workflows? For example, can an agent trained on a demonstration for booking a domestic flight generalize to booking an international one requiring additional steps (passport entry, currency conversion, switching to the international flight tab)?
2. It is unclear how the mouse clicks and keyboard inputs are recorded. What format is it stored in? And does it involve any temporal aspects like waiting for certain elements to load or time-sensitive actions?
3. In Table 1, success rates are reported with different precision for agents (0 dp) and human performance (2 dp). Are they evaluated in the same way?

---

### Official Review · Reviewer_pyP7 · 2025-10-29

**Soundness:** 1
**Presentation:** 2
**Contribution:** 1
**Rating:** 0
**Confidence:** 4

**Summary:**

The paper introduces "Instruction Agent," a training-free framework designed to improve the robustness of GUI agents on complex, long-horizon, or personalized tasks. The core idea is to leverage a single, test-time expert demonstration. This demonstration (a sequence of actions and screenshots) is fed into an "Instructor" module, which uses an LLM (GPT-4O) to generate step-by-step natural language instructions. An "Actor" module then attempts to execute this plan. The Actor is composed of a grounding model (UI-Tars 1.5), an executor (GPT-4O), a "Verifier" (GPT-4O) to confirm if a step was successful by comparing screenshots, and a "Backtracker" (GPT-4O) to attempt recovery from failed steps. The authors evaluate this system on a curated set of 20 tasks from the OSWorld benchmark on which three top-ranked open-source agents fail, reporting a 60% success rate for their method.

**Strengths:**

1. The use of a single, test-time demonstration is a practical and low-cost solution compared to large-scale data collection and model fine-tuning.
2. The paper effectively identifies a significant weakness in current SOTA GUI agents: their brittleness on complex, long-horizon, or idiosyncratic tasks.
3. The inclusion of dedicated Verifier and Backtracker modules is a sensible engineering contribution. The ablation study (Table 2) clearly demonstrates their value, showing that the success rate drops from 60% to 40% when both are removed.

**Weaknesses:**

1. The framework's novelty is somewhat limited. It is primarily an assembly of existing components and concepts: an LLM for plan generation (albeit from a demo), an off-the-shelf grounding model (UI-Tars), and LLM-based modules for visual verification and backtracking. These ideas (e.g., self-correction, reflection) are common in modern agent research. The main contribution is the specific application of these components to a test-time demonstration, but the underlying techniques are not new.
2. The experimental validation is a significant concern:
- Single Benchmark: The evaluation is confined to a single benchmark (OSWorld). Claims of general utility would be much stronger if the framework were validated on other complex GUI environments (e.g., WebArena, Windows Agent Arena).
- Small and Curated Task Set: The evaluation is performed on a random sample of 20 tasks. This number is too small to draw robust statistical conclusions; the 60% success rate (12 tasks) is subject to very high variance.
- Uninformative Baseline Comparison: The 0% success rate for baselines (Table 1) is a result of the task selection criteria, not a fair head-to-head comparison. The tasks were pre-selected specifically because these agents failed them. This setup inflates the perceived contribution of the Instruction Agent and makes the 60% success rate difficult to interpret relative to the benchmark-wide scores (e.g., 42.9% for Computer Use Agent O3 model) mentioned in the introduction.

3. The paper's presentation needs polishing. The citations in the bibliography are poorly formatted.

**Questions:**

1. The paper states the 20 tasks were randomly sampled from a pool of 130 failed tasks. Does the 0% baseline success rate in Table 1 refer to the entire 130-task pool, or did you re-run the baselines on this specific 20-task sample and confirm they all failed?
2. Following the ablation study, what is the success rate of a "vanilla" agent (e.g., just UI-Tars + Executor) when given the perfect instructions? This would help disentangle the 40% "base" performance (from the instructions) from the +20% lift provided by the Verifier and Backtracker.

---

### Official Review · Reviewer_MYB2 · 2025-10-30

**Soundness:** 4
**Presentation:** 4
**Contribution:** 4
**Rating:** 4
**Confidence:** 4

**Summary:**

The paper introduces the **Instruction Agent**, a GUI agent designed to automate complex tasks by leveraging expert demonstrations. Unlike current agents that struggle with novel UI elements, long-horizon actions, and interruptions, the Instruction Agent extracts step-by-step instructions from a single demonstration and strictly follows the user's intended trajectory. It uses **verifier** and **backtracker** modules to handle unexpected interruptions and assess outcomes during execution. Experimental results in the OSWorld environment show that the Instruction Agent achieves a 60% success rate on tasks that other top-ranked agents fail to complete, providing a reliable and practical solution for real-world GUI task automation.

**Strengths:**

## Strengths

- **One Demonstration Suffices**: Unlike methods that require large datasets for model training, the Instruction Agent only needs a single demonstration to automate a new task, significantly lowering the usage threshold and enabling even non-expert users to easily "teach" the AI to complete specific tasks.

- **Outstanding Performance**: Achieves a 60% success rate on difficult tasks that other top agents fail to complete, showcasing its robustness, reliability, and practical applicability.

**Weaknesses:**

## Weaknesses

- **Limited Novelty in the "Instructor-Executor" Framework**: The approach used in the paper is similar to imitation learning, where expert trajectories are segmented and then replayed for action execution. The low-level controller is based on a planner, which is not updated. This method is akin to techniques used in Voyager[1] and Cradle[2], with the addition of providing a successful trajectory as a condition.

- **Insufficient Comparison with Strong Baselines**: The paper lacks comparisons with key baselines, such as MLLM agents with memory input. A classic comparison could involve whether such agents (e.g., RAP[3]) can accomplish similar tasks. In the multimodal retrieval field, success trajectories have been used to improve MLLM agent performance on long and complex tasks, which should have been explored.

- **Limited Experimental Environments**: The OSWorld tasks are relatively simple and can often be described with language, lacking the complexity needed to fully support the paper's conclusions (e.g., line 104). The paper should be tested on more challenging and complex benchmarks, such as Mobile-Bench-v2[4] and ANDROIDWORLD[5], to validate the agent's robustness in a broader range of environments.

[1]Wang, Guanzhi, et al. "Voyager: An open-ended embodied agent with large language models." arXiv preprint arXiv:2305.16291 (2023).

[2] Tan, Weihao, et al. "Cradle: Empowering foundation agents towards general computer control." arXiv preprint arXiv:2403.03186 (2024).

[3]Kagaya, Tomoyuki, et al. "Rap: Retrieval-augmented planning with contextual memory for multimodal llm agents." arXiv preprint arXiv:2402.03610 (2024).

[4]Xu, Weikai, et al. "Mobile-Bench-v2: A More Realistic and Comprehensive Benchmark for VLM-based Mobile Agents." arXiv preprint arXiv:2505.11891 (2025).

[5]Rawles, Christopher, et al. "Androidworld: A dynamic benchmarking environment for autonomous agents." arXiv preprint arXiv:2405.14573 (2024).

**Questions:**

See weakness above.

---

### Official Review · Reviewer_2eQH · 2025-11-01

**Soundness:** 2
**Presentation:** 1
**Contribution:** 2
**Rating:** 2
**Confidence:** 3

**Summary:**

The paper introduces a GUI agent framework. It consists of an Instructor that extracts stepwise instructions from human demonstrations, and an Actor that executes the task strictly following those instructions. To enhance robustness, the system further integrates a Verifier (which checks success) and a Backtracker (which reverts and retries upon failure). On a set of 20 OSWorld tasks where all sota open-source agents fail, the proposed approach achieves a 60% success rate, with ablation studies demonstrating the contributions of the Verifier and Backtracker.

**Strengths:**

- The proposed approach enables reproduction of long, multi-step workflows from a single demonstration, which is relevant to real-world office automation.
- Achieves 60% success on the challenging subset of OSWorld tasks where all state-of-the-art open agents fail, with ablation studies quantifying the benefits of the Verifier and Backtracker.

**Weaknesses:**

- Experiments are limited to 20 tasks where competing methods all fail, which may introduce task selection bias. Broader evaluations (including solvable tasks) are missing. With only 20 tasks, the statistical robustness is weak, and no variance or confidence intervals are reported.
- The approach relies on a GPT-4o + UI-TARS 1.5 pipeline. Its generalization to other LLMs or grounding models, as well as the trade-offs in cost and latency, are not discussed.
- The work does not compare against “demonstration-based imitation/replay/retrieval-augmented” methods.
- References are incorrectly formatted throughout. Many of them should use \citep. Please revise all citations accordingly and recheck for typos after the fixes.

**Questions:**

- Why was the assessment restricted to the 20 “failure” tasks instead of the broader OSWorld or Windows Agent Arena benchmarks? Could the authors share the full task list?
- What is the average token count per instruction, latency per task, and total inference cost? How much time savings are achieved compared with direct human execution?
- Can a single demonstration transfer to slightly modified UIs (e.g., different themes, resolutions, or versions)?
- Are recall/precision or false-positive/false-negative metrics available for the Verifier?
- How would performance and cost change when replacing GPT-4o with a smaller local LLM?

---

### Note · Authors · 2025-11-12

I have read and agree with the venue's withdrawal policy on behalf of myself and my co-authors.